

# Application of visual communication in digital animation advertising design using convolutional neural networks and big data

Jingyi Fang and Xiang Gong

Department of Arts Design, Visual Communication Design Major, Dankook University, Yongin-si, Korea

## ABSTRACT

In the age of big data, visual communication has emerged as a critical means of engaging with customers. Among multiple modes of visual communication, digital animation advertising is an exceptionally potent tool. Advertisers can create lively and compelling ads by harnessing the power of digital animation technology. This article proposes a multimodal visual communication system (MVCS) model based on multimodal video emotion analysis. This model automatically adjusts video content and playback mode according to users' emotions and interests, achieving more personalized video communication. The MVCS model analyses videos from multiple dimensions, such as vision, sound, and text, by training on a large-scale video dataset. We employ convolutional neural networks to extract the visual features of videos, while the audio and text features are extracted and analyzed for emotions using recurrent neural networks. By integrating feature information, the MVCS model can dynamically adjust the video's playback mode based on users' emotions and interaction behaviours, which increases its playback volume. We conducted a satisfaction survey on 106 digitally corrected ads created using the MVCS method to evaluate our approach's effectiveness. Results showed that 92.6% of users expressed satisfaction with the adjusted ads, indicating the MVCS model's efficacy in enhancing digital ad design effectiveness.

## INTRODUCTION

Visual communication, enabled by the advent of big data, represents a burgeoning technological frontier. It has emerged as a critical means of information transmission, and how to achieve more targeted information transmission, especially in digital advertising, remains a topic of great interest to researchers. Big data technology can potentially improve the efficiency of digital advertising by analyzing and processing large amounts of data (*Chen & Zeng, 2021*) in a way that is easy to comprehend (*Kim & Lee, 2021*). However, the efficacy of this technology is hampered by the limited availability of training data. Big data can remedy this limitation. Sophisticated data analysis techniques enable creators to analyze vast data and convey data characteristics, trends, and correlations

Corresponding author
Jingyi Fang, fjy0621@163.com

through images or charts (*Liu et al., 2021*). Combining visual communication with big data can hasten the reader's acquisition of data details. Through a diverse array of chart forms, data features and patterns can be swiftly discovered, leading to in-depth analysis, problem identification, and practical solutions (*Rahaman et al., 2021*). Big data-based visual communication technology is widely employed across various industries, such as business, finance, medicine, and education, where it plays a crucial role. It helps companies understand data, grasp market trends, formulate effective strategies, and boost competitiveness (*Zhang & Tao, 2021*). Therefore, big data-based visual communication technology is an indispensable tool that can effectively convey complex information, help readers understand, enhance the effective utilization of data, and provide robust support for business development. The proposed MVCS model in this article aims at emotion recognition and accurately estimates audio and video multimodal data.

The MVCS model adopts an innovative multimodal emotion analysis approach, harnessing audio and visual data. Initially, the audio data feature extraction subnetwork and video data feature extraction subnetwork receives the speech emotion feature map and the video character information data map, respectively, and extract the low-level features of the data. Depth-wise separable convolution combined with residual modules is utilized to learn standard and higher-level feature information. Subsequently, the two feature maps are concatenated in the depth direction. The concatenated feature map enters the second feature fusion subnetwork. It repeatedly learns its standard features and higher-level feature information through depth-wise separable convolution kernels and residual modules based on big data's mixed knowledge of prior information. Incorporating an attention mechanism and knowledge graph in the MVCS model enables the model to accurately identify and comprehend emotional information in multimodal data.

Additionally, the MVCS model can adaptively adjust model parameters to address various emotion recognition tasks, thus improving the model's generalization ability and accuracy (*Li et al., 2021*). Moreover, the MVCS model can dynamically adjust the playback of the video based on the user's emotional state and interaction behaviour, including scrolling speed, colour contrast, and single-frame clipping, and can provide real-time suggestions for adjusting scrolling speed, contrast, and whether to clip single frames (*Zhang et al., 2021*). Extensive tests were conducted on several datasets, including CMU-MOSI, AffectNet, MSP-IMPROV, and MELD, demonstrating that the F1 value of the MVCS model surpassed that of similar methods. Satisfaction tests were also performed on digitally corrected ads pushed to users using the MVCS model, revealing that 92.6% of users were content with the results. This indicates that the MVCS model can effectively enhance the design of digital ads. Overall, applying the MVCS model in the domain of multimodal emotion recognition presents promising prospects and could provide novel opportunities for advancing artificial intelligence technology. Overall, using the MVCS model in multimodal emotion recognition has broad prospects and can bring new possibilities for the development of artificial intelligence technology.

## RELATED RESEARCH

### Construction of performance management index system

With the rapid development of digital media, the importance of visual communication in marketing is increasing. Animated advertising, as a new promotional tool, has been recognized by increasingly more advertising practitioners for its flexible and varied forms of expression and creative features. Compared with traditional print advertising, animated advertising has higher interactivity and emotional appeal. Using rich graphic elements, vivid visual effects, and humour, animated advertising can attract audience attention, convey product information, and guide consumers to make purchase decisions. In addition, animated advertising has better memorability and impact, which can be deeply imprinted in consumers' minds and sustainably promote brand marketing effects. Therefore, with the growing demand for video content, animated advertising will become a promising marketing method (*Lu et al., 2021*).

At the same time, with the continuous development of technology, the production of animated advertising is becoming increasingly convenient and straightforward, and many online platforms and software can help advertisers quickly produce lively, engaging, and brand-specific animated advertising. This trend reduces the production cost of animated advertising and provides more opportunities for companies to showcase their brands and products. Therefore, the role of visual communication in animated advertising will continue to strengthen in the future and become an indispensable part of the advertising marketing field. How to achieve more effective visual communication has always been a hot research topic.

The utilization of big data technology for visual communication has emerged as a new trend due to its rapid development and application (*Xu et al., 2021*). This technology has the potential to assist companies in understanding and using big data to improve their competitiveness by fulfilling market demands. Additionally, the visualization of complex data through big data visual communication technology can facilitate comprehension of key data points and features (*Gao et al., 2021*). In 2020, *Yu et al. (2021)* explored using big data and visual communication for bimodal emotion calculation, which is expected to achieve more effective emotion prediction. *Li et al. (2017)* proposed an extensive data-based advertising recommendation system in 2017. It analyzed and processed advertising evaluation data and presented the results through heat maps, scatter plots, and other visualizations, allowing users to select the ads they were interested in more easily. In 2021, *Zhang, Gao & Sun (2021)* utilized big data to address the limitations of unimodal emotion recognition. With the development of deep learning, researchers have begun to focus more on multimodal emotion calculation. For instance, *Yang (2022)* conducted the first study on emotion recognition based on audio and video. They separately extracted emotional features from audio and video and integrated them through a feature layer.

The integrated emotional features were then sent to the classifier for emotion recognition (*Ding & Li, 2021*). Their experimental results showed that the results of the two modalities were much better than those of any single mode. They used information from the audio mode and text mode for emotion recognition. These studies indicate that

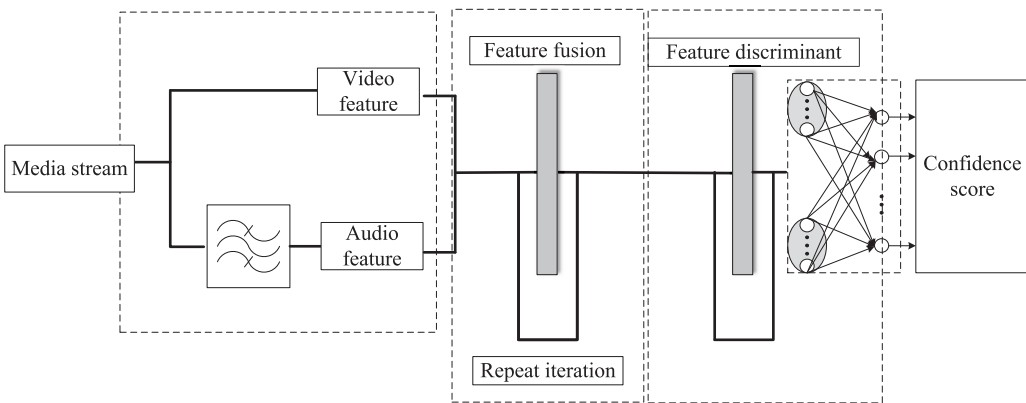

**Figure 1 The overall framework of the traditional multimodal sentiment analysis algorithm.**

using big data combined with visual communication can effectively analyze and display complex information, help users better understand and use data, and achieve better decision-making and prediction results. However, these methods are inefficient in using big data and cannot output visualization suggestions with generalization significance during analysis, and there is limited research on improving user satisfaction. The traditional multimodal emotion recognition architecture is shown in Fig. 1. This architecture is mainly composed of multiple modules. First, the audio and video data are processed by their feature extraction modules to extract low-level feature information and fused in the feature fusion module to obtain higher-level feature information. Next, the classifier classifies this high-level feature information to get emotional prediction results. The traditional multimodal emotion recognition framework has many disadvantages, such as high complexity and low efficiency. New methods and techniques are needed to improve significant data-based multimodal emotion recognition efficiency and effectiveness.

In this article, a pre-trained model was stored and fast gradient convergence of the pre-trained model was implemented using MLP. The output model was then inputted as the initialization feature for the network. We used an MLP with 32 hidden layers as a feature extractor. We used knowledge distillation to shrink the large data Date to speed up the training of large models. To make the pre-trained MLP converge faster, we used positional encoding. Then, we input the video and audio information into a multimodal processing network for processing and finally output the predicted F1 value. We also selected frames with the maximum single-frame emotional fluctuation, the entire emotional fluctuation playback speed, and the screen's maximum emotional fluctuation colour contrast and provided specific suggestions.

# MVCS MODEL FRAME

## Knowledge distillation and pre-training in big data modeling

The system comprises three components: input, convolutional processing, and output. The input component is divided into two processing branches for images and audio. Images are compressed into NumPy arrays through pyramidal compression, while the audio branch

processes input through tensors before being fed into the convolutional processing component. The convolutional processing component performs feature extraction using a distilled model trained on large datasets, yielding multimodal classification results *via* the output component. Knowledge distillation in big data aims to improve a model's generalization performance and efficiency by transferring knowledge from a large, high-performance model to a small, lightweight model. To achieve feature transfer, we use VGG and RESNET as feature extraction networks and use the large-scale video dataset YouTube-8M, open-sourced by Google, as the primary dataset for feature extraction. The YouTube-8M dataset contains millions of labelled video segments from YouTube videos. It is designed to help researchers develop better video understanding and analysis algorithms and provide basic data for training deep learning models. With this dataset, we have sufficient data to perform reliable pre-training.

## Audio and video data processing

We performed frontend data processing and feature extraction to implement sentiment analysis functionality for processing input video streams. In terms of feature extraction, we need to extract emotion-related features from the video for subsequent sentiment classification. In this study, we used both audio and video modalities for sentiment analysis. Specifically, we used convolutional neural networks (CNNs) and extended short-term memory networks (LSTMs) to extract features from audio and video, respectively. Then we fused the features of the two modalities. For audio data, we used one-dimensional CNNs to perform convolution operations on the time-domain signals. For video data, we used three-dimensional CNNs to perform convolution operations on the spatial-temporal signals. After feature extraction, we concatenated the features of the two modalities along the channel dimension and fed them into a fully connected layer for sentiment classification. Our experimental results show that the multimodal sentiment analysis method using feature fusion has significantly improved accuracy compared to single-modal and simple feature concatenation methods. This indicates that feature fusion is an effective multimodal sentiment analysis method. We used a pre-trained and optimized VGG16 network and a Resnet50 network with the best learning rate adjusted.

Our BiLSTM network consists of two LSTM layers in opposite directions, one scanning the sequence from the beginning to the end and the other scanning it from the end to the beginning. The outputs of these two LSTM layers are concatenated to form the final output. The calculation formula is as follows (*Xie & Xie, 2021*):

$$i_t = \sigma(W_{xi}x_t + W_{hi}h_{t-1} + b_i)$$
$$f_t = \sigma(W_{xf}x_t + W_{hf}h_{t-1} + b_f) \tag{1}$$
$$o_t = \sigma(W_{xo}x_t + W_{ho}h_{t-1} + b_o)$$

Including the calculations of the forget gate, output gate, and input gate, the formula for calculating the candidate cell is as follows (*Abdar, Nikou & Gandomi, 2021*):

$$\tilde{c}_t = \tanh(W_{xc}x_t + W_{hc}h_{t-1} + b_c) \tag{2}$$

We set the parameters of the BILSTM network as follows: the hidden layer dimension is 256, the sequence length is 1,024, the batch size is 1, and the initial learning rate is 1.2. We used the Adadelta algorithm for training, with a training step of 1,000 for each epoch. To prevent overfitting, we also added a dropout layer with a dropout probability of 0.5. At the same time, we adopted the early stopping strategy to avoid overfitting and make the training process more efficient. We set an early stopping mechanism to prevent overfitting and save time and computing resources. In addition, we also saved the model parameters with the highest accuracy on the validation set for future use.

## Multimodal subspace correlation propagation

We must calculate their spatial transfer matrix to achieve the subspace transfer of multimodal feature vectors (*Ko, Ko & Lin, 2019*). In this article, we define a unified relationship matrix $UMR_{(LMR)}$ for multiple modalities in a video as follows:

$$\mathbf{L}_{urm} = \begin{vmatrix} \lambda_{11}L_{image} & \lambda_{12}L_{i-a} & \lambda_{13}L_{i-t} & \lambda_{14}L_{i-s} \\ \lambda_{21}L_{a-i} & \lambda_{22}L_{ondio} & \lambda_{23}L_{s-t} & \lambda_{24}\mathbf{L}_{a-s} \\ \lambda_{31}L_{t-i} & \lambda_{32}L_{t-a} & \lambda_{33}L_{text} & \lambda_{34}L_{t-s} \\ \lambda_{41}L_{s-i} & \lambda_{42}L_{s-a} & \lambda_{43}L_{s-1} & \lambda_{44}\mathbf{L}_{shot} \end{vmatrix} \tag{3}$$

After defining the unified relationship matrix Lurm, we must consider setting the initial values. Similarity calculation within the same modality is relatively easy. Limage and Laudio are calculated based on Euclidean distance, and LTE can be computed using cosine distance. For Li-S, La-S, and Lt-S, we set the initial correlation between the image, audio, or text and their corresponding shots to one and all others to 0, *i.e.*, the identity matrix. Similarly, Lshot is also initially set to the identity matrix. For the initial correlation between different modalities, *i.e.*, Li-a, Li-t, and La-t, we use the method of co-occurrence data embedding (CODE) to calculate them.

Assuming the prior probability distributions of two categorical variables X and Y are $\bar{p}(x,y)$, The idea of the CODE algorithm is to use their statistical relationship to embed two categorical variables X and Y into a low-dimensional Euclidean space *via* two mappings $\varphi : X \rightarrow \mathbb{R}^d, \psi : Y \rightarrow \mathbb{R}^d$ Embed them into a low-dimensional Euclidean space. The model is established as follows:

$$\mathbf{S}_{usm} = \begin{vmatrix} s_{11} & s_{12} & \cdots & s_{1T} \\ s_{21} & s_{22} & \cdots & s_{2T} \\ \vdots & \vdots & \ddots & \vdots \\ s_{1T} & s_{2T} & \cdots & s_{TT} \end{vmatrix} \tag{4}$$

Among them, each element Sa, Sb, and Sc of Susm represents the similarity relationship of each data object (in this article, namely image, audio, and text) in a unified subspace. T is the total number of data objects in the unified subspace, *i.e.*, T = 3 * N. It should be noted that the arrangement order of each data object in Lurm and Susm should be consistent. Therefore, as discussed in SimFusion, we have the following:

$$\mathbf{S}_{usm}^{new} = \mathbf{L}_{urm}\mathbf{S}_{usm}^{original}\mathbf{L}_{urm}^{T} \tag{5}$$

This is just the most fundamental similarity enhancement calculation in SimFusion. We can continue to iterate the calculation until convergence or until we obtain relatively satisfactory results.

$$\mathbf{S}_{usm}^{n} = \mathbf{L}_{urm}\mathbf{S}_{usm}^{n-1}\mathbf{L}_{urm}^{T} = \mathbf{L}_{urm}^{n}\mathbf{S}_{usm}^{0}\left(\mathbf{L}_{umm}^{T}\right)^{n} \tag{6}$$

After iterative calculations, we can obtain the final result.

$$\mathbf{S}_{usm}^{final} = \begin{vmatrix} \mathbf{S}_{image} & \mathbf{S}_{i-a} & \mathbf{S}_{i-t} & \mathbf{S}_{i-s} \\ \mathbf{S}_{a-i} & \mathbf{S}_{audio} & \mathbf{S}_{a-t} & \mathbf{S}_{a-s} \\ \mathbf{S}_{t-i} & \mathbf{S}_{t-a} & \mathbf{S}_{text} & \mathbf{S}_{t-s} \\ \mathbf{S}_{s-i} & \mathbf{S}_{s-a} & \mathbf{S}_{s-1} & \mathbf{S}_{shoo} \end{vmatrix} \tag{7}$$

In this way, by mapping the original data to a low-dimensional semantic subspace through LPP, while denoising and discovering intrinsic features, the coordinates in the semantic subspace can represent the original data, which serves as one of the input conditions for SVM, namely, the feature vector of training data. After large-scale data distillation and pre-training, we can input the video stream into the improved MVCS model.

## MVCS backend processing network

From Fig. 2, the process of the MVCS model is an end-to-end process. The video is first separated into two parts, audio and video, and input into a set of deep networks. The network consists of $3 \times 3$ depth-wise convolution, $1 \times 1$ point-wise convolution, batch normalization (BN), ReLU activation function, and residual stacking (*Vivekananda & Khapre, 2021*). At the same time, the total number of frames, contrast curve, and playback speed of the video is also calculated. Then, the video stream is input into a pre-trained feature convolutional network comprising 256 hidden layers. The hyperparameter $\gamma$ is a training parameter extracted after distillation, and the pre-trained weight value $\alpha$ 0 is added to output the network structure's feature maps jointly. In the final part of the feature discrimination subnet, the joint data feature map is transformed into a 2,048-dimensional feature vector using a depth-wise separable convolution kernel with a residual module. Since the feature extraction network "feature fusion" in this part has already obtained the primary emotional feature data, it can be used for model evaluation. Therefore, it can generate emotional fluctuations for single frames and emotional fluctuation values under different playback speeds and extract the attention softmax to input it into the attention-based feature enhancement network. At the same time, the maximum emotional fluctuation K value for a single frame, emotional fluctuation M value under different playback speeds, and emotional fluctuation E value under different resolutions are also extracted, and the mean square error between the three feature values and the predicted values in the base model is summarized to provide improvement suggestions (*Liu, Chen & Tian, 2021*; *Zhi, Zhou & Li, 2021*).

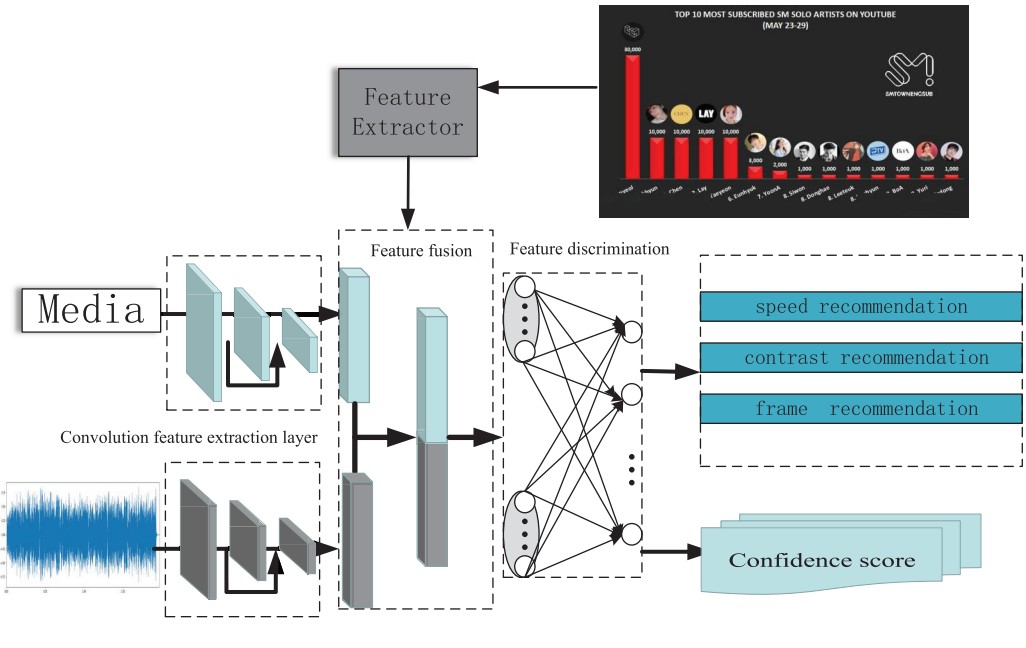

**Figure 2  Model framework of MVCS network structure.**

# EXPERIMENT

## Experimental environment and data sets

The dataset used for model training is the publicly available Google dataset YouTube-8M. The test datasets are four publicly available audio and video emotion recognition datasets: CMU-MOSI, AffectNet, MSP-IMPROV, and MELD. This article has experimentally validated all single-modal, multimodal emotion recognition tasks. All experiments in this article were run on the laboratory server. The server development platform is the Linux operating system Ubuntu 18.04, with an NVIDIA 2080Ti graphics card, and the development language is Python. The deep learning framework used is Keras, with a TensorFlow backend. The dataset used in the experiment is the MOST (*CVPR, 2019*).

The optimizer used for training is stochastic gradient descent (SGD), with a momentum of 0.9, an initial learning rate of 0.045, and a learning rate decay rate of 6% every two cycles. In addition, Polyak averaging is used to create the final model used for testing.

## Dual-modal dataset recognition results

During the experiment, the CMU-MOSI dataset was split into training, validation, and testing sets at a ratio of 6:2:2. The model was trained on the training set. The training process was observed using the validation and the experimental results were obtained on the testing set. For the AffectNet dataset, only the video part was used. Since the dataset did not provide a division of training/validation/testing sets, we used the standard cross-validation method for the experiment. Specifically, we divided the dataset into five parts, used one part as the validation set and the rest as the training set for each experiment, and then averaged the results of the five experiments. For the MSP-IMPROV dataset, we
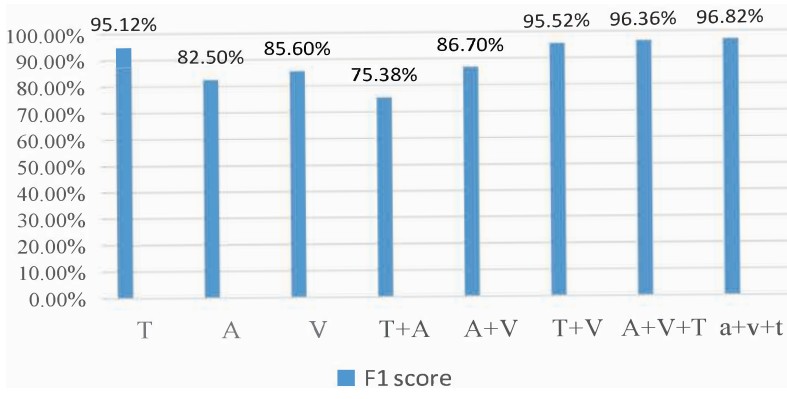

**Figure 3 Accuracy of the binary classification experiment.**

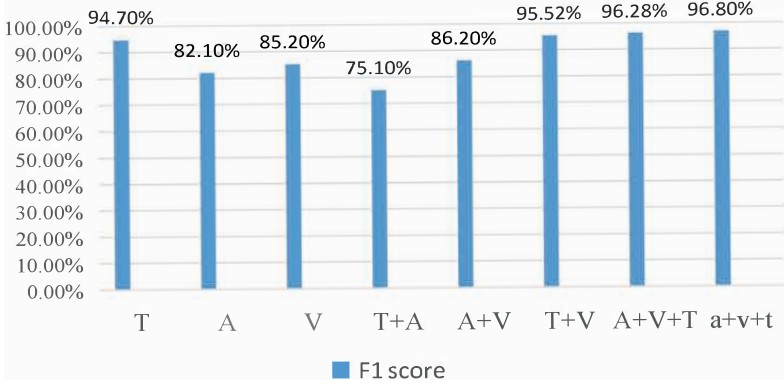

**Figure 4 Fl score.**

divided it into training and testing sets. In particular, we divided the data into two parts for our experiment: the first 3 months were used as the training set and the last 3 months as the testing set. The model was trained on the training set and evaluated on the testing set. Figures 3 and 4 present the comparison of binary classification accuracy and F1 score, respectively.

T represents the text modality. A represents the audio modality, V represents the video modality, T+A represents the fusion of text and audio modalities, T+V represents the fusion of text and video modalities, A+V represents the fusion of audio and video modalities, A&V+T(Con) represents the directly cascaded multimodal fusion method, and a+v+t(Att) represents the multimodal fusion method with an attention mechanism. We conducted experiments on single-modality, double-modality, and multimodality emotion recognition tasks, proving that the proposed model performed well on different tasks [26].

We conducted a comprehensive analysis of the phenomenon and concluded that the lack of feature similarity between voice and video modes might be the cause. To address this issue, we propose utilizing more advanced models or more effective feature extraction methods to improve the fusion effect of voice and video modes. In the final group of experiments, we introduced the fusion method of attention mechanism into the BiLSTM

network, which resulted in the best recognition effect, with a recognition accuracy of 76.82% and an Fl value of 76.8%.

## Three-mode high-dimensional audio and video emotion analysis

In this study, we conducted three-mode high-dimensional audio and video emotion analysis using the DCCA algorithm to orthogonalize the features of multiple modes and express them in the same space for neural network classification. The results showed that extracting high-level audio signal features from the 3D feature map with static MFCC information and higher-order differential information is feasible using the BiLSTM network structure. This is likely due to the BiLSTM network's full utilization of its feature learning ability, with each layer having a larger receptive field than the previous layer. By stacking multiple convolutional layers, the network can extract hierarchical semantic features from the original audio signals, achieving the task of audio emotion analysis. Moreover, this method can also reduce the network's dependence on high-level features, improving its generalization ability. The experimental results demonstrated that our proposed method outperformed traditional video feature extraction methods and was more discriminative in emotion recognition tasks.

Additionally, the multimodal method proposed in this article performed better in emotion recognition tasks than other single-mode emotion analysis methods. Our stacked convolution neural network also demonstrated advantages in audio signal processing by understanding the characteristics of audio signals from different levels and modelling more extended time series, capturing more semantic information. This provides a valuable reference for future audio emotion analysis tasks.

To assess the efficacy of the fusion model introduced in this study, we juxtaposed it with four earlier fusion algorithms: feature level fusion, decision level fusion, model level fusion, and the most recent convolution neural network fusion model. Figure 5 shows that our multimodal fusion model outperformed the other four fusion algorithms on three datasets, attesting to its effectiveness in emotion recognition tasks. Our proposed model achieved the optimal performance on both bimodal and three-mode emotion recognition tasks on the CMU-MOSI dataset and the three-mode emotion recognition task on the AffectNet dataset. Our model excelled in the bimodal emotion recognition task on the MELD dataset and achieved the most exceptional results on the three-mode emotion recognition task. This reveals that our proposed multimodal fusion model manifests good generalization performance on different datasets and can be employed for various emotional recognition tasks.

Moreover, we conducted additional experiments to investigate the influence of various modalities on the efficacy of our proposed model. The outcomes revealed that each modality had a distinct impact on emotional recognition performance, and integrating multiple modalities could significantly enhance recognition accuracy. Specifically, visual modality greatly influenced the emotional recognition task more than auditory and text modalities. Nonetheless, amalgamating all three modalities yielded optimal performance, underscoring the significance of multimodal fusion in emotional recognition.

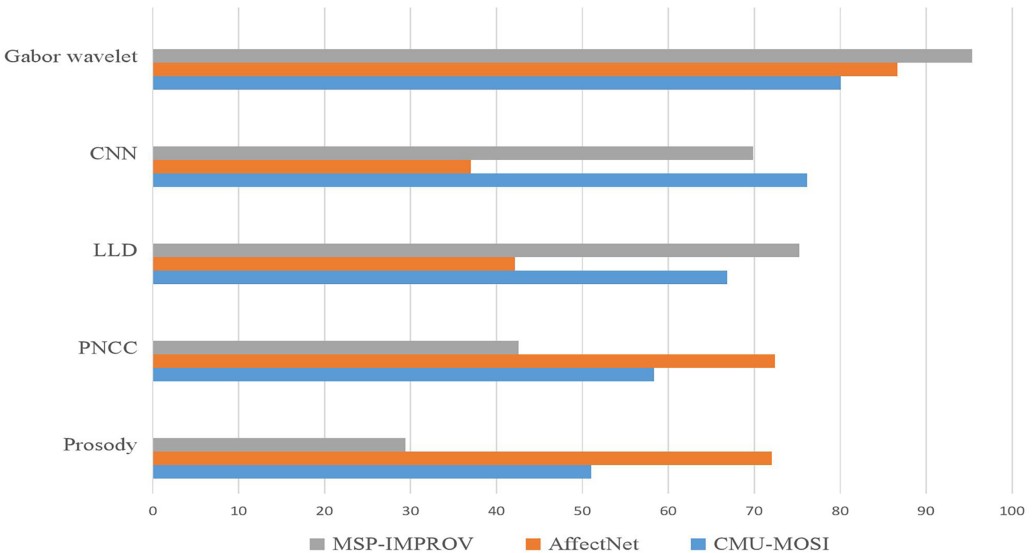

**Figure 5 Satisfaction and accuracy of the improvement scheme on the EffectNet dataset.**

Additionally, we performed an ablation analysis to scrutinize the efficacy of individual components of our proposed multimodal fusion model. Our findings indicated that every component, such as feature extraction, modality attention, and modality fusion, played a crucial role in enhancing the model's overall performance. The modality attention mechanism aided in concentrating on the most informative modality, whereas the modality fusion technique proficiently integrated information from diverse modalities, resulting in superior performance.

Overall, our proposed multimodal fusion model showed promising results in emotion recognition tasks and had the potential to be applied in various practical applications such as affective computing, human-computer interaction, and multimedia analysis. Future work could further explore combining additional modalities, such as physiological signals or facial expressions, to improve the model's performance.

## Cross-media recommendation results

In visual communication-based advertising design, emotional analysis can significantly enhance advertisements' communication effectiveness while alleviating the creators' workload.

To gauge the efficacy of video cross-modal recommendation, this experiment leverages the multimodal characteristics of video to recommend advertisements. Specifically, advertisements that may pique a user's interest are recommended based on their score record on the KDD Cup advertising dataset. Table 1 displays the map and NDCG@k values, respectively, indicating accuracy and satisfaction. MVCS outperforms other methods regarding recommendation performance in both tables, while the results recommended by cmLDA are second only to the methods proposed in this study.

Based on the user's rating history of the advertising dataset, we utilized a cross-modal recommendation algorithm to suggest movie posters and photos that may interest the user.

**Table 1 Recommended mAP and NDCG@k value.**

| MAP | mmLDA | corrLDA | cmLDA | MVCS | MDL+KNN |
| --- | --- | --- | --- | --- | --- |
| Top5 | 0.54 | 0.70 | 0.67 | 0.086 | 0.58 |
| Top10 | 0.78 | 0.88 | 0.61 | 0.098 | 0.61 |
| Top20 | 0.61 | 0.76 | 0.54 | 0.092 | 0.59 |
| NDCG@k | mmLDA | corrLDA | cmLDA | MVCS | MDL+KNN |
| K = 5 | 0.57 | 0.60 | 0.68 | 0.076 | 0.64 |
| K = 10 | 0.49 | 0.56 | 0.61 | 0.069 | 0.61 |
| K = 20 | 0.49 | 0.65 | 0.52 | 0.073 | 0.59 |

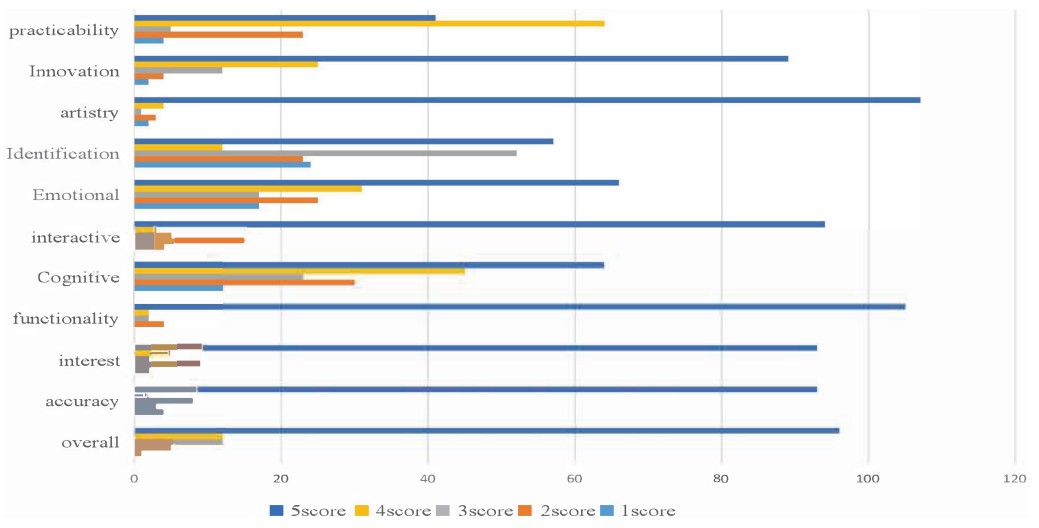

**Figure 6 Effect picture of user rating satisfaction.**

Half of the user-rated data was used as known information to sort the movie-related images and other images in the database for recommendations. To evaluate the algorithm's effectiveness, we conducted a crowd satisfaction survey using several advertisements processed by multimodal emotional analysis, depicted in Fig. 6. The survey results indicate that our proposed MVCS algorithm achieved remarkable performance in recommendation accuracy, enabling advertising users to achieve precise targeting. The table shows that the overall user satisfaction rate was 92.6%.

## CONCLUSION

In the current era of big data, effective advertising content delivery requires integrating multiple modes, such as vision and hearing. Using big data, such as customer data and market trends, can facilitate the creation of more targeted advertising. Improvement in video communication quality and user interest in videos is crucial for enhancing the efficiency and accuracy of visual communication. This article proposes a framework, MVCS, that employs multimodal emotion analysis to enhance video recommendation methods through semi-supervised learning. MVCS leverages multimodal content analysis

to describe user characteristics from multiple modes, effectively addressing the problem of overly simplistic recommendation results in single-mode recommendations and sparse user data. The experimental analysis validates the proposed method's effectiveness in improving the recommendation system's performance and overcoming the problem of sparse user ratings on standard data sets. MVCS has demonstrated favourable results across multiple data sets, indicating its potential to enhance video transmission quality and user experience. The experimental model shows that the MVCS framework achieves an accuracy of 81.51%, 83.47%, 72.04%, and 82.44% in predicting user sentiment on the CMU-MOSI, AffectNet, MSP-IMPROV, and MELD data sets, respectively, with corresponding adjustment suggestions generated. The system accuracy surpasses 90%, higher than comparable methods, with an overall user satisfaction rate of 93%, enabling effective content push.

In the era of big data, effective advertising requires utilizing multiple forms of communication, such as visual and auditory. To create more targeted ads, leveraging big data sources such as customer data and market trends is crucial. Improving video quality and user engagement is vital in enhancing visual communication efficiency and accuracy. This article proposes a multimodal sentiment analysis framework called MVCS, which improves video recommendation methods through semi-supervised learning, enhancing the accuracy and completeness of content recommendations. By utilizing multimodal content analysis, MVCS can describe user characteristics from multiple perspectives, addressing the issue of overly simplistic recommendations resulting from single-modal analysis and sparse user data. Through experiments on standard datasets, the proposed method effectively improves recommendation performance and overcomes the problem of sparse user ratings. MVCS performs well on multiple datasets, demonstrating its potential to improve video transmission quality and user experience. The experimental model achieves accuracies of 81.51%, 83.47%, 72.04%, and 82.44% in predicting user emotions for the CMU-MOSI, AffectNet, MSP-IMPROV, and MELD datasets, respectively, and generates corresponding adjustment suggestions. The system's accuracy exceeds 90%, higher than comparable methods, achieving effective content delivery with an overall user satisfaction rate of 93%. The proposed MVCS model can be extended to other visual communication fields, aiding meaningful work in areas such as medical education, and has broad application prospects. In today's big data era, effective advertising requires harnessing multiple modes of communication, such as sight and sound. Leveraging big data sources like customer data and market trends is essential to create more targeted advertising. Improving the quality of video communication and user engagement with video is also vital to enhancing the efficiency and accuracy of visual communication. This article proposes a multimodal emotion analysis framework called MVCS, which improves video recommendation methods using semi-supervised learning. By leveraging multimodal content analysis, MVCS can describe user characteristics from multiple perspectives, addressing the problem of overly simplistic recommendations from single-mode analysis and sparse user data. Through experiments on standard datasets, the proposed method is effective at improving recommendation performance and overcoming

the problem of sparse user ratings. MVCS also performs well across multiple datasets, demonstrating its potential to enhance video transmission quality and user experience. The experimental model achieves an accuracy of 81.51%, 83.47%, 72.04%, and 82.44% in predicting user sentiment on the CMU-MOSI, AffectNet, MSP-IMPROV, and MELD datasets, respectively, with corresponding adjustment suggestions generated. The system accuracy surpasses 90%, higher than comparable methods, with an overall user satisfaction rate of 93%, enabling effective content delivery.

### Funding
The authors received no funding for this work.

### Competing Interests
The authors declare that they have no competing interests.

### Author Contributions
- Jingyi Fang conceived and designed the experiments, performed the experiments, analyzed the data, performed the computation work, prepared figures and/or tables, authored or reviewed drafts of the article, and approved the final draft.
- Xiang Gong conceived and designed the experiments, performed the experiments, analyzed the data, performed the computation work, prepared figures and/or tables, authored or reviewed drafts of the article, and approved the final draft.

### Data Availability
The code is available in the Supplemental Files.

The data is available at YouTube: 230K human-verified segment labels, 1,000 classes, 5 segments/video, https://research.google.com/youtube8m/download.html.

### Supplemental Information
Supplemental information for this article can be found online at http://dx.doi.org/10.7717/peerj-cs.1383#supplemental-information.

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
