# Peer review of "Application of visual communication in digital animation advertising design using convolutional neural networks and big data"

_PeerJ Computer Science, doi:10.7717/peerj-cs.1383_

## Round 0.1 · original submission · Major Revisions

Dear colleagues

Your manuscript has potential but it needs some major improvements as suggested by the experts in the field. Please carefully revise the technical stance and non technical stance eg improve the language of the manuscript professionally.

Then you can resubmit the updated article.

·

Basic reporting

The CONTRIBUTION, MOTIVATION and ORIGINALITY need to be further addressed. Moreover, GRAMMAR CORRECTION procedure need a particular focus.

Experimental design

An overview is required at the beginning of Section 3 to help readers better understand the construction ideas of the model. Moreover, some sentences are very difficult to understand. “In this paper, we stored the grid model catch.pt through pre-training and implemented rapid gradient convergence of the pre-trained model using MLP, and input the output model as the network's initialization feature.”

Validity of the findings

I am contented with the validity of the related findings in this work.

Additional comments

By training large-scale video data sets, MVCS model can analyze video from multiple dimensions such as visual, sound and text. The convolutional neural network is used to extract the visual features of the video, and the cyclic neural network is used to extract the features and analyze the emotion of audio and text. By integrating the feature information, the video playback mode can be dynamically adjusted according to the user's emotional state and interactive behavior, and the amount of advertising can be increased. This is a good research paper for the present perspective. You all had a great effort overall. I really appreciate it. Please follow the following comments for improving the quality of this article.

1. The major contributions and innovations of this research remain unclear.
2. I'm unsure of the contribution of this work overall - the motivation of the article should be constructed according to this comment. Please make it clear.
3. The originality of the study and its contribution to the literature have not been adequately addressed.
4. The title of the article needs to be revised,try that the titles and abstracts should have some reflection of computer science related terms;
5. An overview is required at the beginning of Section 3 to help readers better understand the construction ideas of the model;
6. Some sentences are very difficult to understand. “In this paper, we stored the grid model catch.pt through pre-training and implemented rapid gradient convergence of the pre-trained model using MLP, and input the output model as the network's initialization feature.”
7. Generally speaking, MLP is a general concept. Which method does it specifically refer to in the passage?
8. The authors used VGG and RESNET as feature extraction networks, but it doesn't explain how they were optimized.
9. For non-original descriptions, more references should be added, such as formula (1) and formula (2).

·

Basic reporting

This paper proposes a multi-modal sentiment analysis framework MVCS, which enhances video recommendation by semi-supervision. Based on multi-modal content analysis, users' characteristics are described from multiple modes, which can effectively make up for the over-simple recommendation results in the single-mode recommendation and solve the problem of sparse user data. Through the experimental analysis, the experimental results on the standard data set verify that the proposed method can effectively solve the problem of sparse user ratings and improve the performance of the recommendation system. The authors have presented an interesting idea in the paper. However, I have a few suggestions for the authors to incorporate in the revised draft.
(1) The English language should be improved by professional organizations or native English speakers.
(2) In the abstract/conclusion, readers prefer to understand the application scenarios and social value of the research;
(3) The number of references is not enough, which reflects that the author does not have a good summary of multi-modal deep learning methods.
(4) Add the number of citations to 25+, avoiding the use of Chinese literature;

Experimental design

(1) The distribution of paragraphs in each chapter is still not reasonable, and the author needs to correctly distinguish the relationship between research methods and research objectives;
(2) What is "grid model Catch-pt"?

Validity of the findings

(1) Theoretical background needs to be reviewed more rigorously and deeply, for example, I liked to see theoretical background of identification of emotional states and interactive behaviors
(2) Please make sure your 'conclusion' section underscore the scientific value added of your paper, and/or the applicability of your findings/results, as indicated previously. Please revise your conclusion part into more details.

---

## Round 0.2 · accepted · Accept

Dear authors thanks for incorporating the reviewers suggestions, they are now satisfied with the current version of the paper. Good luck for your future research.

·

Basic reporting

No comment

Experimental design

No comment

Validity of the findings

No comment

Additional comments

The authors have satisfactorily resolved all of my concerns. Therefore, I suggest this article for publication in its current form.

·

Basic reporting

The updated version is looking good.

Experimental design

Authors have incorporated my comments.

Validity of the findings

Looking good.